# Digital Twin: Origin to Future

Maulshree Singh [1], Evert Fuenmayor [1], Eoin P. Hinchy [2], Yuansong Qiao [3], Niall Murray [3] and Declan Devine [1,*]

1    Material Research Institute, Athlone Institute of Technology, Athlone, Co., N37 HD68 Westmeath, Ireland; m.singh@research.ait.ie (M.S.); efuenmayor@ait.ie (E.F.)
2    Confirm Centre for Smart Manufacturing, School of Engineering University of Limerick, Limerick, Co., V94 T9PX Limerick, Ireland; Eoin.Hinchy@ul.ie
3    Software Research Institute, Athlone Institute of Technology, Athlone, Co., N37 HD68 Westmeath, Ireland; yuangsongqiao@ait.ie (Y.Q.); nmurray@ait.ie (N.M.)
*    Correspondence: ddevine@ait.ie; Tel.: +353-879-187-012

**Abstract:** Digital Twin (DT) refers to the virtual copy or model of any physical entity (physical twin) both of which are interconnected via exchange of data in real time. Conceptually, a DT mimics the state of its physical twin in real time and vice versa. Application of DT includes real-time monitoring, designing/planning, optimization, maintenance, remote access, etc. Its implementation is expected to grow exponentially in the coming decades. The advent of Industry 4.0 has brought complex industrial systems that are more autonomous, smart, and highly interconnected. These systems generate considerable amounts of data useful for several applications such as improving performance, predictive maintenance, training, etc. A sudden influx in the number of publications related to 'Digital Twin' has led to confusion between different terminologies related to the digitalization of industries. Another problem that has arisen due to the growing popularity of DT is a lack of consensus on the description of DT as well as so many different types of DT, which adds to the confusion. This paper intends to consolidate the different types of DT and different definitions of DT throughout the literature for easy identification of DT from the rest of the complimentary terms such as 'product avatar', 'digital thread', 'digital model', and 'digital shadow'. The paper looks at the concept of DT since its inception to its predicted future to realize the value it can bring to certain sectors. Understanding the characteristics and types of DT while weighing its pros and cons is essential for any researcher, business, or sector before investing in the technology.

**Keywords:** digital twin; Industry 4.0; digital model; system optimization; predictive maintenance

## 1. Introduction

In the era of Industry 4.0, when several industries are experiencing a digital transformation, Digital Twin (DT) is considered no less than a linchpin for gaining a competitive and economic advantage over competitors. DT saw its origins in the aerospace industry, and it is expected to revolutionize other industries [1]. The main applications of DT in different sectors are designing/planning, optimization, maintenance, safety, decision making, remote access, and training, among others. It can be a great tool for companies to increase their competitiveness, productivity, and efficiency [2]. DT has the ability to link physical and virtual worlds in real time, which provides more a realistic and holistic measurement of unforeseen and unpredictable scenarios [3].

The value DT brings to any sector, by reducing time to market, optimizing operations, reducing maintenance cost, increasing user engagement, and fusing information technologies, is indisputable [4]. The global market of DT was estimated at USD 3.1 billion in 2020 [5] and is expected to grow exponentially in succeeding years. The outbreak of the COVID-19 pandemic has changed the way production and maintenance are looked at, which has accelerated the adoption of Digital Twins [6]. Thus, it becomes essential to

understand what the implications of DT implementations could or should be, depending on the nature of the industry where it is adopted.

This paper provides a detailed account of different types of DT and their characteristics. In literature, DT is often described and reviewed within a manufacturing context [2,7–14], whereas this review looks at DT as a technology beyond just manufacturing as well as consolidates different types of DTs at one place, which has not been reported before. This work aims to provide an overview of the current state-of-the-art of DT as well as a classification system of what is to be considered DT based on key required functionalities. The paper intends to analyze the role and advantages of DT and how can it bring value to the existing systems. Academia defines DT in several different ways depending on their field of research; therefore, this paper will also look at the origin and different definitions of DT throughout the literature to create a clearer picture of what can be referred to as DT and what should not be. The paper proposes a consolidated definition of DT that can be used irrespective of any industry and its application.

The paper is divided into four sections. Section 2 explains the origins of DT and its early applications while also exploring its definitions throughout the literature. Section 3 investigates the advantages and characteristics of DT and how it has been classified into different categories based on different criteria by different authors. The future applications of DT and the challenges associated with it are discussed in Section 4 of the paper.

## 2. Digital Twin in Literature

### 2.1. History of Digital Twin

Even though DT technology has gained massive popularity in the past couple of years, the concept is not entirely new. Its concept came into being in relation to Product Lifecycle Management (PLM) in 2002 at the University of Michigan by Michael Grieves [15]. The proposed model has three components: real space, virtual space, and linking mechanism for the flow of data/information between the two; the model was then referred to as 'Mirrored Spaces Model' [16]. A similar concept in which software models mimic reality from information input from the physical world was imagined by David Gelernter in 1991 and was called 'Mirror Worlds' [17]. In 2003, Kary Främling et al. also proposed "an agent-based architecture where each product item has a corresponding 'virtual counterpart' or agent associated with it" as a solution to the inefficiency of transfer of production information via paper for PLM [18]. By 2006, the name of the conceptual model proposed by Grieves was changed from 'Mirrored Spaces Model' to 'Information Mirroring Model' [15,19]. The model put emphasis on the linking mechanism between two spaces being bidirectional and having multiple virtual spaces for a single real space where alternate ideas or designs can be explored (Figure 1). Due to the limitations of the technologies, such as low computing power, low or no connectivity of devices with the internet, data storage and management, underdeveloped machine algorithms, etc., DT had no practical applications at the time.

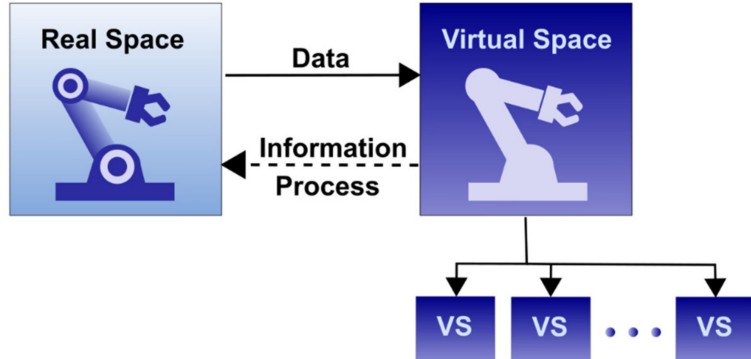

**Figure 1.** Mirrored Spaces Model/Information Mirroring Model as proposed by Michael W. Grieves. (Adapted from [16,19].)

The name 'Digital Twin' (DT) first appears in NASA's draft version of the technological roadmap in 2010 [20]. In the NASA roadmaps, DT was also referred to as 'Virtual Digital Fleet Leader'. NASA was the first association to forge the definition of DT; it was described as "an integrated multi-physics, multi-scale, probabilistic simulation of a vehicle or system that uses the best available physical models, sensor updates, fleet history, etc., to mirror the life of its flying twin". Even though the first mention of DT is in the 2010 roadmap, NASA had used a similar concept before for the Apollo program, where two identical space vehicles were built to mirror each other [21,22]. Soon, the US Air Force followed the footsteps of NASA and used DT technology for the design, maintenance, and prediction of their aircraft [23–25]. The idea was to use DT to simulate physical and mechanical properties of the aircraft to forecast any fatigue or cracks in the structure, thus prolonging the remaining useful life of the aircraft. E. Tuegel [23] and B. T. Gockel et al. [25] in their papers defined DT only for the aircraft and called it 'Airframe Digital Twin' or ADT, which was a computational model to manage the aircraft over its entire lifecycle. Besides monitoring, DT was also proposed for sustainable space exploration as well as for future generations of aerospace vehicles [26]. The timeline of the evolution of DT can be seen in Figure 2.

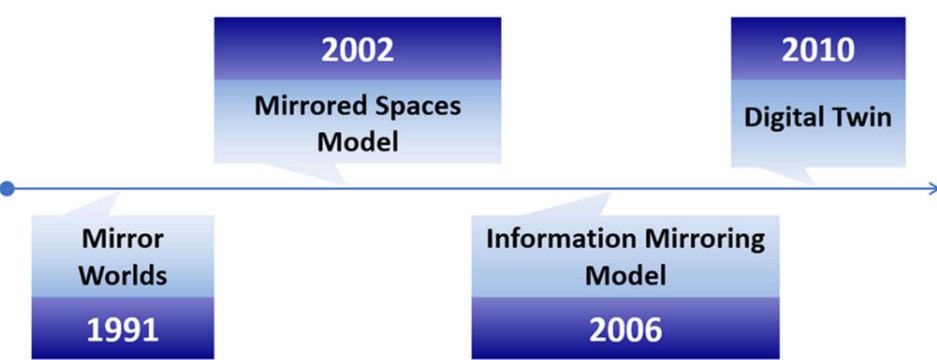

**Figure 2.** Timeline of evolution of Digital Twin.

*2.2. Digital Twin Description in Literature*

Since the first-ever definition published by NASA, different authors have described DT in their own terms and based on its application. The definitions in the literature have referred to DT as a virtual or ***digital model*** [21,23,27–35], ***layout*** [36], ***counterpart*** [7,9,35,37], ***doppelganger*** [38], ***clone*** [39], ***footprint*** [40], ***software analogue*** [41], ***representation*** [42–46], ***information constructs*** [2,47], or ***simulation*** [20,26,48–51] of its physical counterpart. The first few definitions of DT that came out described DT only in respect to the aerospace/aeronautics industry which then ventured into manufacturing, which is why the words such as **'aircraft'** [23,25,30], **'vehicle'** [20,26,48], or **'airframe'** [25,29] were replaced by **'system'** [2,7,21,22,34–36,40,50–52], **'machine'** [28,53], **'product'** [22,32,36,37,44–47,51], **'object'** [9,36,41–43], **'entity'** [2,9], **'asset'** [31,33,39,40], **'device'** [41], or **'process'** [9,21,32,33,36,45,46,52,54]. Ever since DT found its application in industries beyond manufacturing, the definition has also moved from just being restricted to non-living entities or industrial products, and can now be used to describe even complex biological systems as humans and trees, among others [43,55].

One thing that binds most definitions of DT other than being a virtual representation of a physical object is the bidirectional transfer or sharing of data between the physical counterpart and the digital one, including quantitative and qualitative data (related to material, manufacturing, process, etc.), historical data, environmental data, and most importantly, real-time data. Using these data, DT can perform such tasks as:

- In-depth analysis of physical twin;
- Design and validation of new or existing product/process;
- Simulate the health conditions of physical twin;

- Increase safety and reliability of physical twin;
- Optimization of part, product, process, or production line;
- Track the status of physical twin throughout its lifetime;
- Predict the performance of physical twin;
- Real-time control over physical twin.

Definitions of DT tend to overlook its longevity; however, some authors consider DT as a cradle-to-gravel model, meaning that DT can be used over the entire life cycle, from the time of inception of the product until its disposal [23,33,44,50]. However, Grieves and Vickers [47], who conceptualize the idea of DT, have defined a type of DT that is created even before its physical twin exists (see Section 3.3). Martin and Nadja [56] in their review found eleven papers in which DT prequels the physical twin. Grieves and Vickers also suggest that DT technology can have information related to the safe decommissioning of the product during its disposal phase. In addition, after the product is disposed, DT of one generation can help in the design and production of the next generation [57].

It is clear from the literature that DT is different from computer models (CAD/CAE) and simulation. Even though many organizations use the term 'Digital Twin' synonymously to 3D model, a 3D model is only a part of DT [33]. DT uses data to reflect the real world at any given point of time and thus can be used for observing and understanding the performance of the system and for its predictive maintenance [58]. Computer models, just like DT, are also used for the generic understanding of a system or for making generalized predictions, but they are rarely used for accurately representing the status of a system in real time. A lack of real-time data makes these models or simulations static, which means that they do not change or cannot make new predictions unless new information is fed to them [59]. However, having real-time data is not enough for DT to operate—the data also need to be loaded automatically to DT and the flow from physical to digital should be bidirectional. Liu et al. [11] reported that more than half of the papers they reviewed were describing and/or studying 'Digital Model' or 'Digital Shadow' despite authors claiming it to be Digital Twin. W. Kritzinger et al. [2] also found the similar result that there is more literature on 'Digital Model' or 'Digital Shadow' than on 'Digital Twin' (see Section 3.3 for Digital Model' and 'Digital Shadow').

There are other concepts such as 'Product Avatar' or 'Digital Thread' which are also sometimes interchangeably used or get confused with DT. Just like 'Digital Twin', the term 'Digital Thread' was also emanated from the aerospace industry, where it was used to describe an integrated system engineering process used for managing the entire process digitally. It included 3D CAD models, model-based engineering, BoM, manufacturing processes, assembly logistics, delivery systems, etc. [50]. Digital Thread has also been described as the communication framework that consolidates the asset's data and allows seamless data flow; in other words, Digital Thread provides the right information at the right time to the right place [31]. Therefore, as compared to DT, which is a real-time virtual representation of its physical twin, Digital Thread is just the record of the information on the physical twin throughout its lifetime [60]. The major difference between Product Avatar and DT comes from the fact that they have been derived from two different research lines and thus have different capabilities and purpose; a Product Avatar is a digital counterpart of a smart product that lets its user use the attributes and services of that smart product for its entire lifecycle [9].

Due to the presence of a plethora of definitions in the literature, there is no consensus of what can actually be described as DT or not. Thus, giving academics and businesses permission to use the term DT loosely and conveniently to their needs creates confusion between different terminologies related to the digitalization of industries. A sudden influx in the number of publications related to 'Digital Twin' also indicates that interest is increasing exponentially, as it happens with any new promising technology.

In order to simplify the confusion around different terminologies used to describe DT, a definition of DT is proposed here that can be applied irrespective of the industry or its application:

*"A Digital Twin is a dynamic and self-evolving digital/virtual model or simulation of a real-life subject or object (part, machine, process, human, etc.) representing the exact state of its physical twin at any given point of time via exchanging the real-time data as well as keeping the historical data. It is not just the Digital Twin which mimics its physical twin but any changes in the Digital Twin are mimicked by the physical twin too."*

## 3. Advantages, Characteristics and Types of Digital Twin

Since the first introduction of DT, its popularity has increased as more and more researchers started focusing their research on DT. Figure 3 shows the exponential growth of the number of publications found on Scopus as well as ScienceDirect (limited to the English language) that contain the term 'Digital Twin' in the article title, abstract, or as keywords from 2011 to 2020. The increase in the publications is quite recent, i.e., 2016 onwards, as it can be clearly seen in Figure 3.

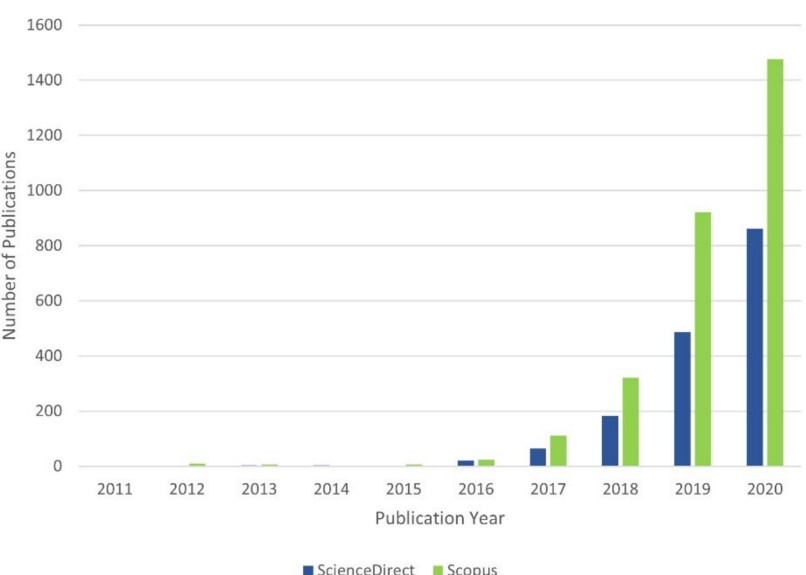

**Figure 3.** Number of Digital Twin-related publications by year from 2011 to 2020 on Scopus and ScienceDirect.

The growing popularity of DT can also be observed as it has been considered as a strategic technology trend for three consecutive years (2017–2019) by Gartner [61–63], future technology in the field of aerospace and defense by Lockheed Martin [64], and one of the defining technology of next decade by Forbes [65]. According to Juniper Research, by 2021, the global DT market was expected to be USD 12.7 billion [66]. The global DT market share, previously dominated by early adopters of the technology i.e., North America and Europe (>50%), is expected to expand in the Asia Pacific at a compound annual growth rate of 40.0% over the next 5 years [67].

As the DT market is growing exponentially, every sector or business wants to invest in the technology. However, to leverage the DT technology for any sector, it is important to understand the value it can bring to the business. It is also beneficial to understand the different characteristics and types of DT in order to choose the correct type of DT that can complement the business and can provide maximum profits.

### 3.1. Advantages of Digital Twin

The main reason DT technology is seen as the cornerstone in Industry 4.0 is its plethora of advantages, including the reduction of errors, uncertainties, inefficiency, and expenses in any system or process. It also removes all the silos in processes or organizations that otherwise work in isolation within compartments and divisions in more traditional industrial structures. Some of the advantages reported for DT include:

- *Speed prototyping as well as product re-designing*: Since simulations allow the investigation of a number of scenarios, the design and analysis cycles shorten, making the whole process of prototyping or re-designing easier and faster. Once implemented, DT can be used in different stages of the product design process, from conceptualizing the idea of the product to its testing [68]. Besides that, it also creates an opportunity where the customization of each product based on users' needs and usage data is possible [69]. Since the DT is connected to its physical twin throughout its lifetime, the comparison between the actual and predicted performance can be made, allowing engineer/product designers to reconsider their assumptions on which the product was designed [70].

- *Cost-effective:* Due to DT involving mostly virtual resources for its creation, the overall cost of prototyping decreases with time. In traditional prototyping, redesigning a product is time-consuming as well as expensive because of the use of physical materials and labour, and on top of that, a destructive test means the end of that costly prototype, whereas using DT, products can be recreated and put through destructive tests without any additional material cost. Thus, assuming even if the cost is equal at the start, the physical costs keep increasing as inflation rises but the virtual cost decreases significantly as time progresses (Figure 4) [47]. DT allows the testing of products under different operating scenarios, including destructive scenarios, without any additional costs. Moreover, DT can reduce operating costs and extend the life of equipment and assets once implemented.

- *Predicting Problems/System Planning:* Using DT, we can predict the problems and errors for future states of its physical twin, providing us an opportunity to plan the systems accordingly. Due to the real-time data flowing between the physical asset and its DT, it can predict problems at different stages of the product lifecycle. This is beneficial especially for products that have multiple parts, complex structures, and are made up of multiple materials such as aircraft, vehicles, factory equipment, etc., because as the complexity of any product increases, it gets harder to predict component failures using conventional methods [68].

- *Optimizing Solutions and Improved Maintenance:* The traditional methods of maintenance are based on heuristic experience and worst-case scenarios rather than on the specific material, structural configuration, and usage of an individual product, making them reactive rather than proactive [26]. However, DT can foresee defects and damage in the manufacturing machine or system and thus can schedule the maintenance of the product in advance. By simulating different scenarios, DT provides the best possible solution or maintenance strategy that makes the maintenance of the product/system much easier. In addition, the constant feedback loop between DT and its physical counterpart can be used to validate and optimize the system's process all the time.

- *Accessibility:* The physical device can be controlled and monitored remotely using its DT. Unlike physical systems, which are restricted by their geographical location, virtual systems such as DT can be widely shared and can be remotely accessed [47]. Remote monitoring and controlling of equipment and systems becomes a necessity in a situation where local access is limited, like during the COVID-19 pandemic when lockdowns have been enforced by governments and working remotely or non-contact is the only viable option [71].

- *Safer than the Physical Counterpart:* In industries such as oil and gas or mining where the working conditions are extreme and hazardous, the capability of DT to remotely access its physical twin, as well as its predictive nature, can reduce the risk of accidents and hazardous failures. However, DT's advantage of accessing remotely is not limited to preventing accidents. During the global COVID-19 pandemic, not having human contact and in-person monitoring is also a way to guarantee safety. According to a recent Gartner survey, almost one-third of companies are using DT amidst the global COVID-19 pandemic to increase the safety of employees and customers through remote monitoring [72].

- *Waste Reduction:* Using DT to simulate and test product or system prototypes in a virtual environment significantly reduces wastage. Prototype designs can be probed and scrutinized virtually, under a variety of different test scenarios, to finalize the final product design prior to manufacture. This not only saves on material wastage but also reduces development costs and time to market.
- *Documentation and Communication:* To create a DT, it is important to synchronize data scattered across different software applications, databases, hard copies, etc., which simplifies the process of accessing and maintaining the data in one place [33]. DT enable a better understanding of system reactions and thus it can be used to document and communicate the behaviour and mechanisms of the physical twin [32].
- *Training:* DT can be used to develop more efficient and illustrative safety training programmes than the traditional one [73]. Before working on a high-risk site or hazardous machinery, operators can be trained using a DT to reduce the dangers, as exposing and educating them about different processes or scenarios will make them confident in dealing with the same situations in person. For example, mining is a high-risk environment where new employees can be trained using DT on machinery operations, as well as how to deal with emergency scenarios [74]. DT can also be a great tool in closing the knowledge gap from experienced workers to newcomers.

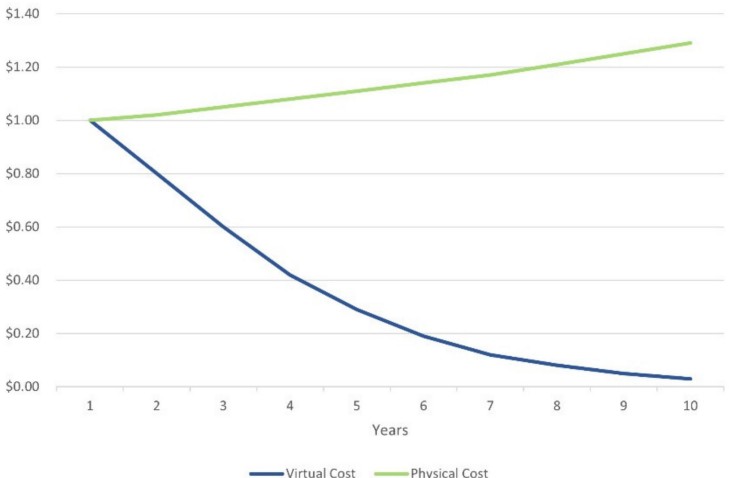

**Figure 4.** Real vs. virtual costs of prototyping. (Adapted from [47].)

*3.2. Characteristics of Digital Twin*

Depending on the type of DT, it can possess distinctive properties from others, but regardless, all DTs have a few characteristics in common:

- *High-fidelity:* A DT needs to be a near-identical copy of its physical counterpart in terms of appearance, contents, functionality, etc., with a very high degree of accuracy. A super-realistic digital model helps DT in mimicking every aspect of its physical twin. Ultra-high fidelity computer models are considered the backbone of the DT [48]. This level of detail allows DT simulation and prediction tools to be more reliable when presented with a set of alternative actions or scenarios [21].
- *Dynamic:* The physical is dynamic, meaning it changes with respect to time. Thus, a DT also needs to change as the physical system changes. This is achieved through the seamless connection and continuous exchange between the physical and virtual worlds. The data exchanging can be dynamic data, historical static data, as well as descriptive static data [9]. DT has been described as a 'living model in 3D' [38]. The objective of DT being dynamic is to mirror the physical twin and its behaviour realistically in the digital world [33].
- *Self-evolving:* A DT evolves along with its physical counterpart throughout its life cycle. Any changes in either the physical or Digital Twin are reflected in its counterpart,

creating a closed feedback loop [33]. A DT is self-adapting and self-optimizing with the help of the data collected by physical twin in real time, thus maturing along with its physical counterpart throughout its lifetime [9].

- *Identifiable:* Every physical asset needs to have its own DT. During different stages of the product lifecycle, the data and information related to it evolve and so does the model, including 3D geometric models, manufacturing models, usage models, functional models, etc. Due to the existence of such models created for DT, a DT can be uniquely identified from its physical twin or vice versa anywhere in the globe and for the entirety of its life cycle [75].
- *Multiscale and Multiphysical:* DT, being the virtual copy of its physical twin, needs to incorporate the properties of the physical twin at multiple scales or levels. Thus, the virtual model in DT is based on macroscopic geometric properties of the physical twin such as shape, size, tolerance, etc., as well as on microscopic properties such as surface roughness, etc. In other words, DT contains the set of information about the physical twin from micro atomic level to macro geometric level [47]. DT is also multiphysical because, besides the aforementioned geometric properties, the model is also based on physical properties of the physical twin such as structural dynamics models, thermodynamic models, stress analysis models, fatigue damage models, and material properties of physical twin such as stiffness, strength, hardness, fatigue strength, etc. [76].
- *Multidisciplinary:* Industry 4.0 revolves around many disciplines, and DT being the backbone of Industry 4.0 sees the fusion of disciplines such as computer science, information technology, and communications; mechanical, electrical, electronic, and mechatronic engineering; automation and industrial engineering; and system integration physics, just to name a few [34].
- *Hierarchical:* The hierarchical nature of DT comes from the fact that the different components and parts that make up the final product all have their corresponding DT model, e.g., DT of an aircraft is comprised of rack DT, DT of the flight control system, DT of the propulsion system, etc. [23]. Therefore, a DT can be seen as a series of integrated submodels [29].

### 3.3. Classification of Digital Twin

Digital Twins (DTs) can be classified into different types based on different criteria such as when the DT is created, level of integration, its applications, hierarchy, and maturity level. Different authors have come up with their own nomenclature of DT types based on these criteria.

#### 3.3.1. DT Creation Time

According to Grieves and Vickers [47], there are two types of DT based on when is it developed during the life cycle of the product—before the prototype is created, i.e., at the designing phase, or after the product is ready, i.e., at the production phase. Both types of DTs are integrated and operated for multiple usages on a platform they called Digital Twin Environment (DTE).

- *Digital Twin Prototype (DTP):* DTP can be described as a DT that contains the set of data/information that is essential to create or manufacture a physical copy from the virtual version. This includes BOM (bill of materials), design files, CAD models, etc. The product cycle will start from the creation of DTP, which can be put through several tests, even the destructive ones, before creating its physical twin. In addition, DTP helps us in identifying and avoiding unpredictable and undesirable scenarios that are difficult to identify with traditional prototyping. Once DTP is complete and validated, its physical twin can be manufactured in the real world. The accuracy of the simulation/model will determine the quality of the physical twin.
- *Digital Twin Instance (DTI):* This type of DT is connected to its physical counterpart throughout its life cycle. DTI came into being during the production phase. Once

a physical system has been built, the data from the real space are sent to the virtual space and vice versa to monitor and predict the system behaviour. With these data, it can be found out if the system is depicting the predicted desirable behaviour or not, as well as if the predicted undesirable scenarios has been successfully eliminated. Since the linkage between both the systems is bidirectional, any changes in one will be duplicated on the other. A collection of DTIs is called Digital Twin Aggregate (DTA) by the authors.

### 3.3.2. Level of Integration

Based on the integration level of DTs, Kritzinger et al. [2] divided them into three subcategories (Figure 5):

- *Digital Model:* In this type of DT, the data between the physical and digital object are exchanged manually, due to which any changes in the state of the physical object are not reflected in the digital one directly, and vice versa.
- *Digital Shadow:* The data from the physical object flow to the digital automatically, but this is still manual the other way around. As a result, any change in the physical object can be seen in its digital copy, but not vice versa.
- *Digital Twin:* In this type of DT, there is an automatic bidirectional flow of data between the physical and digital object. Therefore, the changes in either object, physical or digital, directly lead to changes in the other.

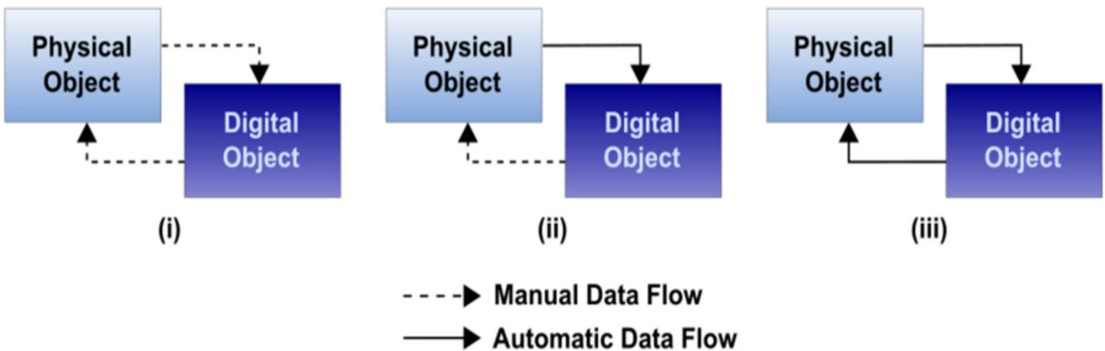

**Figure 5.** Types of DT based on level of integration (adapted from [2]). (**i**) Digital Model; (**ii**) Digital shadow/Static Digital model; (**iii**) Digital Twin/Dynamic Digital Model.

Centre for Digital Built Britain [77] describes Digital Shadow as 'Static Digital Model' and Digital Twin as 'Dynamic Digital Model' as the data in the dynamic one are fed live, which can be used for a real-time control mechanism, whereas in the static model, data are added with time, making it more suitable for strategic planning.

### 3.3.3. Application

DT can also be categorized depending on their applications. The two broad applications of a DT are prediction and interrogation [47]. A **Predictive DT**, as the name suggests, predicts future behaviour and performance of its physical counterpart whereas an **Interrogative DT** is used to interrogate the current or past state of its physical counterpart irrespective of its location. DTs can also be divided depending on if the focus of application is on product, process, or performance [46,78]:

- *Product DT:* It is used for prototyping as it analyses the product under different conditions and makes sure that the next physical product is behaving as planned. By virtually validating the product, the prototyping can be rapid as the total development time is reduced and there is no longer a need to develop multiple of them.
- *Production DT:* It is used for validating the processes by simulating and then analyzing them even before the actual production. This helps in developing an efficient

production methodology under different conditions. The data from Product and Production DT can be used together for monitoring and maintenance of the machinery.

- *Performance DT:* It is used for decision-making processes by capturing, Aggregating, and analyzing data from smart products and plants. Since Performance DT includes performances of both product and production, it optimizes the operations depending on the availability of plant resources, which creates an opportunity to improve on the Production and Product DTs via a feedback loop.

### 3.3.4. Hierarchy

From a hierarchal perspective, DT can be divided into three different levels as well (Figure 6), according to the magnitude involved in manufacturing [75]:

- *Unit level:* It is the smallest participating unit in manufacturing and can be a piece of equipment, material, or environmental factors. Unit-level DT is based on the geometric, functional, behavioural, and operational model of unit-level physical twin.
- *System level:* It is an amalgamation of several unit-level DTs in a production system such as production line, shop floor, factory, etc. Interconnectivity and collaboration among multiple unit-level DT lead to a wider flow of data and better resource allocation. A complex product, e.g., aircraft, can also be considered as system-level DT.
- *System of Systems (SoS) level*: A number of system-level DT are connected together to form SoS-level DT, which helps in collaborating different enterprises or different departments with an enterprise such as supply chain, design, service, maintenance, etc. In other words, SoS-level DT integrates different phases of the product throughout its life cycle.

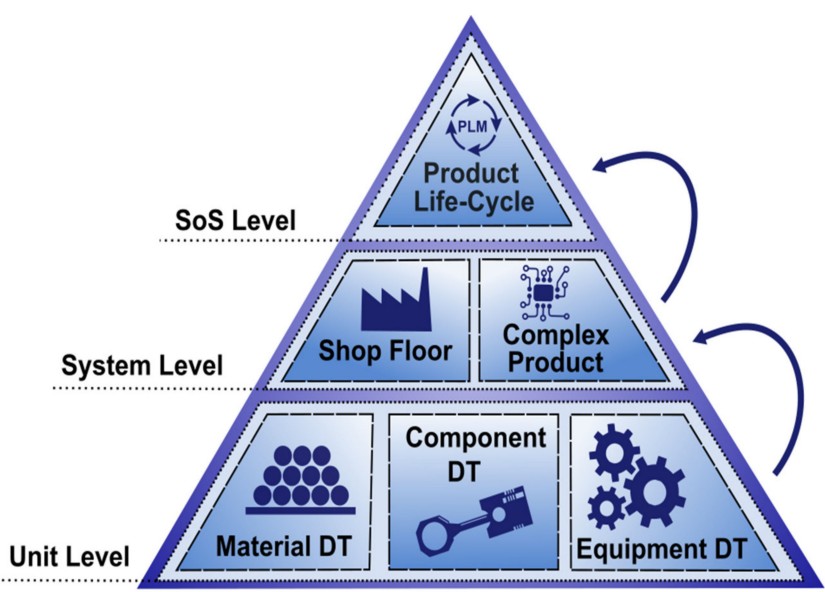

**Figure 6.** Hierarchical levels of DT in manufacturing. (Adapted from [75].)

The hierarchy of DT can also be classified as *(i) Part/Component twin, (ii) Product/Asset twin, (iii) System twin, and (iv) Process twin* [79,80], with part twin being the simplest. More sophisticated and comprehensive systems/processes can be achieved by putting together the lower-level twins.

### 3.3.5. Level of Maturity/Sophistication

Based on the sophistication level of DT, i.e., the quantity and quality of data obtained from the physical twin and its environment, DTs can be grouped into [81]:

- *Partial DT:* It contains a small number of data points, e.g., pressure, temperature, humidity, etc., which is useful in determining the connectivity and functionality of DT.

- *Clone DT:* It contains all significant and relevant data from the product/system that can be used for making prototypes and categorizing development phases.
- *Augmented DT:* It utilizes data from the asset along with its historical data and at the same time derives and correlates the useful data using algorithms and analytics.

The sophistication level of DT can be improved with the accumulation of bigger sets of data over times of operation. For Azad M. Madni et al. [58], the level of maturity of DT is not just limited to data but it also encompasses the sophistication level of the virtual representation/model. On this basis, DT is divided into four levels:

- *Pre-Digital Twin:* This is level 1, where DT is created prior to the physical asset for the purpose of making decisions on prototype designs to reduce any technical risk and resolve issues upfront by using a generic system model.
- *Digital Twin:* Level 2 incorporates the data from the physical asset related to its performance, health, and maintenance. The virtual system model uses these data to assist high-level decision-making in the design and development of the asset, along with scheduling maintenance. The data transfer at this level is bidirectional.
- *Adaptive Digital Twin:* This level 3 provides an adaptive user interface between physical and Digital Twin, and has the capability to learn from the preferences and priorities of human operators using supervised machine learning. Using this DT, real-time planning and decision-making during operations is possible.
- *Intelligent Digital Twin:* In addition to the features from level 3, level 4 has unsupervised machine-learning capability, making it more autonomous than level 3. It can recognize patterns in the operational environment and using that along with reinforced learning allows for a more precise and efficient analysis of the system.

## 4. Future and Challenges of Digital Twin

The technologies composing DT such as Internet of Things (IoT), Industrial Internet of Things (IIoT), Artificial Intelligence (AI), big data, simulation, and cloud computing, among others, have been on a path of constant evolution, and thus it can be assumed that DT will continue to evolve in parallel to these technologies. Therefore, the true potential of DT is still hidden behind the fog of its future form. This is evident by the estimates about the DT market worldwide. The global market of DT is expected to grow at the rate of 58% and in just six years, i.e., by 2026, it is expected to reach USD 48.2 billion [5]. The COVID-19 pandemic is one of the key factors driving the growth of the DT market as it is expected that post-pandemic, industry will be pushing for the further digitization of processes.

Humans are an integral part of any industry and need to be taken into account when developing any DT. IEEE (The Institute of Electrical and Electronics Engineers) believes that in the future, DT will become an indispensable part of machine development as well as of the human–machine symbiotic relationship [82]. Some researchers see 'Digital Triplet' as the next evolution of DT [83–85]. In addition to physical and cyber worlds, Digital Triplet also includes the 'intelligent activity world' where humans will solve problems by using DT. Unlike DT, where systems and process are automated, in Digital Triplet human interaction with the system and process is also accounted for, thus creating value from data using human intelligence and knowledge. The aim of 'Digital Triplet' is to support engineering activities throughout a product's life cycle including design, manufacturing, use, maintenance, remanufacturing, and circulation of resources by integrating all three worlds, just like DT [83,84].

### 4.1. Future Applications of Digital Twin

DT technology has been leveraged in both the aerospace and manufacturing sectors, but there are several other sectors where DT is still in its infancy, such as agriculture, construction, automobile, and healthcare. One of the driving factors pushing the demand for DT in several sectors is the COVID-19 pandemic in 2020. The demand for DT has increased in healthcare and pharmaceutical industries as well as in manufacturing industries due to this viral outbreak [5].

In manufacturing, DT is expected to become a central capability in MBSE (model-based systems engineering) in the future, the reason being that using DT MBSE can be applied to the entire life cycle of the system. DT can help MBSE in penetrating new markets such as manufacturing, construction, and real estate [58]. DT is expected to improve future non-destructive testing by integrating it into the manufacturing process through online monitoring and make it capable of decision-making [86]. In the future, it can be possible that the manufactured products will have their own DTs and we will see more cases of connected DTs.

In the automotive industry, General Electric [87] believes that with advances in predictive maintenance and analytics, DT technology will be widely implemented throughout the industry, from individual car owners to manufacturers. For car owners, the maintenance of the car will become a lot easier as unexpected breakdowns will be a thing of the past and the car will be able to book maintenance appointments itself. The mechanics/service industry will be able to provide faster and more efficient solutions as they will have all the right information about the car at their fingertips even before the appointment. Besides managing their clients, DT will help garages in managing their inventory and supply chain more efficiently. As for car dealers, DT can take care of monitoring the performance and health of the entire car fleet by itself along with helping in making crucial business and financial decisions based on depreciation and asset performance management. All the data created by the future cars and garages can be analyzed, improving the entire automotive value chain. IBM (International Business Machines Corporation) [38] too claims that in coming years, DT technology will save considerable amounts of time and money in the automotive industry, as DT will be built long before the cars hit the assembly line. This will provide a data pool on motor type, suspension, chassis, aerodynamics, and even different types of drivers who will interact with the vehicle, enabling designers and engineers to design and model an ideal product and also to observe and analyze the performance of the vehicle even before it hits the road. DT technology can also be extended to motorbikes, where if a bike's part breaks then instead of ordering a generic replacement part, the bike's DT can order a tailor-made, generatively designed replacement part that can be made using 3D printing technology [88].

The global COVID-19 pandemic has given DT a platform to grow in the healthcare industry. The authors who developed 'Cardio Twin' believe that their platform can prevent Ischemic heart disease (IHD) and stroke in the future [89]. As DT technology advances in the medical field, we can expect that every person on the planet will be able to receive extremely personalized medical treatment and care in cost-effective way [90].

DT is in its early stages of development in the agriculture industry. DT can be exploited in farming for (1) storing and collecting data, (2) categorizing actions in complex workflow, (3) automated data analysis, (4) learning and measuring the content and capacity of the soil, (5) simulating the crop outcome, (6) predicting the weather, and (7) recognizing the stress on resources by factors such as invasive plants and animals, soil quality, pollution, etc. [91].

As space technology advances, it will become important for engineers and scientists to make these explorations more sustainable so that we can explore further into the universe. The Commonwealth Scientific and Industrial Research Organization (CSIRO) [92], Australia's national science agency, which has been working on DT solutions for mining operations, is envisioning to build DTs to aid mining processes in future space exploration by collaborating with the Australian and international space agencies to make space operations more sustainable.

More and more national and regional governments are gaining interest in creating DTs of their own cities. These cities with their own DT will be better planned and managed. The administration will be able to serve its stakeholders better and resolve the problems faster. These cities' DTs have added advantage in cases of natural disasters such as earthquakes, cyclones, and tsunamis, among others. DTs of cities can help in cases of emergency

such as global pandemics, as multiple cities will be able to share the local strategy and status via DT in a timely manner [93].

In future, DT technology can be combined with other technologies for better results, e.g., mixed reality with DT technology can be used for better visualization in DT as well as for remotely supervised inspections [86]. Besides that, DT is finding its application in the education sector. Sepasgoza [94] showed how combining DT with augmented and virtual reality (AR/VR) can be used for developing digital pedagogy for architecture, engineering, and construction students, which can be valuable for educators delivering remote classes, either in emergencies such as pandemics or as a part of regular online-based learning.

### 4.2. Challenges to Implement Digital Twin

Realizing a mammoth technology like DT comes with its own challenges. The challenges that arise with developing a DT depends on its scale and complexity but there are some barriers with the technology that are common to all. DT technology being in its infancy stages means that even though it has great potential, its implementation carries complications that can be either engineering and technology-related or can be commercial [95]. Such complications include ambiguity surrounding the definition or concept of DT, lack of appropriate tools, expensive investments, data-related issues, lack of rules and regulations, etc. The reader, in the following paragraphs, can find a compilation of the most common themes currently preventing a streamlined adaptation by industries of DT technologies.

#### 4.2.1. Novelty of Technology

As DT is still an emerging technology, there is a lack of clear understanding about the value it can bring to individuals, businesses, or industries. Incompetency on the part of technical and practical knowledge is also hindering the progress of the technology. There is also a lack of case studies of successful practices or business models implementing DT into company activities or realistic estimations on the costs involved in this implementation [96].

Several technologies come together to make DT a reality such as 3D simulations, IoT/IIoT, AI, big data, machine learning, and cloud computing. Since these technologies themselves are in developing phases, it impedes the evolution of DT. The infrastructure to implement DT needs to be improved to enhance the efficacy of the technology. There is a need for further research in technologies such as high-performance computing technology, machine learning technology, real-time virtual-real interactive technology, intelligent perception and connection technology, among others, in order to implement DT [68].

Along with the infrastructure, there is a need for supporting software. There are a plethora of software packages providing DT solutions such as Predix by General Electronics (GE), Azure Digital Twin by Microsoft, PTC, 3D Experience by Dassault Systèmes, ABB LTD, Watson by IBM, Digital Enterprise Suite by Siemens, etc. [97], which makes it harder to identify and chose the platform that can deliver the most appropriate service based on the specific needs of the interested industries/business.

#### 4.2.2. Time and Cost

One of the biggest challenges DT needs to overcome to reach its full potential is the high cost associated with its implementation. The whole process of developing ultra-high-fidelity computer models and its simulation of processes to create a DT is a time-consuming and labour-intensive exercise that also requires a huge amount of computational power to run, thus making DT an expensive investment [49]. On top of that, embedding the existing system with sensors for data collection along with the requirement of high-performance IT infrastructure, which includes hardware as well as software for storing and processing that data, contribute to the additional cost. Gartner analyst and expert Marc Halpern has also shown concern over cost and time-related aspects of DT at the PDT Europe conference in Gothenburg, Sweden, saying that bringing DT concepts together can take more time and resources than one can imagine [98]. A paper published by West and Blackburn gives a

glimpse of the scale of cost and time invested in bringing DT into reality. The authors claim that it could cost trillions of dollars and hundreds of years to completely implement Digital Threads/Digital Twins of weapons systems for the U.S. Air Force, making it impractical to fully realize the technology [99]. This makes it very crucial for industries to perform cost-benefit analysis before implementing DT.

### 4.2.3. Lack of Standards and Regulations

As there are a plethora of DT models and architectures available in the literature, there is a need for defining a consistent framework for DT throughout the industry that includes shared and mutual understanding of interfaces and standardization for uniformity together with efficient design of data flow to make accessibility of data easier without compromising its security [100]. Standardizing models, interfaces, protocols, and data is essential for efficient third-party communication, product and human safety, data security, and integrity, especially in industries such as aerospace, automobile, healthcare, etc. [95]. Besides that, standards and standards-based interoperability need to be developed to address the social and organizational challenges unfolded by digital transformation within industries [101]. A lack of device communication and data collection standards can compromise the quality of data being processed for DT, which will be reflected on its performance [102]. Since the technologies including big data and AI are also still in their infancy, the laws and regulations around them are yet to be formalized. There is a standardized framework ISO 23247 (Digital Twin Manufacturing Framework) under development which is aimed at providing guidelines and methods for developing and implementing Digital Twins in the manufacturing sector. This framework will have four parts: (i) Overview and general principles, (ii) Reference architecture, (iii) Digital representation of physical manufacturing elements, and (iv) Information exchange [103].

### 4.2.4. Data Related Issues

As DT technology deals with the data, one of the biggest concerns that arises is about privacy, confidentiality, transparency, and ownership of these data.

Owning and sharing data is influenced by company policies as well as by the mindset of people and society about data ownership, thus putting a limitation on DT that is beyond the complexities of technology and engineering [95]. Not having proper policies in place regarding sharing the data internally (within the organization) or externally (stakeholders across the supply chain) can lead to data silos within different departments of an organization, which can be detrimental to the value chain [95] as data silos lead to inconsistency and synchronization issues [104]. Another possible issue that needs to be considered is how to share the data among different DTs, i.e., data interoperability. In a setting where there are multiple DTs at different hierarchical levels, each generating a different type of data and one feeding on the other DT can create a complicated relationship between data set that causes data interoperability issues [105]. Cybersecurity cannot be neglected when it comes to handling the data. On one hand, where having data silos can affect the overall performance of DT, not having them makes DT more vulnerable to cybercrime [105].

Another major challenge regarding data involved in DT is its convergence and governance. Projects involving big data are likely to fail due to lack of data governance and management to tackle the challenges related to big data, which include identifying and accessing data, transforming data from different sources, poor quality of data, translation loss, etc. [104].

### 4.2.5. Life-Cycle Mismatching

An additional concern over DT technology is related to the products that have long life cycles such as buildings, aircraft, ships, machinery, or even cities. The life cycles of such products are far longer than the validity of the software used for designing or simulating the DT as well as for storing and analyzing the data for DT [106]. This means that there

is a high risk, in the future at some point of time, of either the formats used by software becoming obsolete or becoming locked with the same vendor for new versions of software or other authoring tools [105,107].

## 5. Conclusions

Though the concept of DT is decades old, its real impact has only come into being in recent years. The digital market is and will continue to grow as more and more end-use industries adapt the technology owing to its potential in reducing operational costs and time, increasing the productivity of the existing system, improving maintenance, easing accessibility, creating a safer work environment, and other purposes yet to be realized.

DT technology, when combined with the other technologies such as AR/VR, mixed reality, additive manufacturing, 3D printing, etc., as well as with other DTs, will open doors for new applications and potential. Although this technology comes with its own challenges, the benefits are far greater. This paper identified different types of DTs to better understand and distinguish what can and cannot be described as DT to edge closer to consensus on its definition and limitations. Identifying and understanding the potential of DT in any sector and complementing it with the pertinent type of DT provides an opportunity for developing Industry 4.0 tools that offer numerous advantages, from simulation and prediction capabilities to record-keeping and forensic troubleshooting. This paper looked at the advantages offered by implementing this technology and contrasted them with the challenges. The benefits involve reductions in cost, increased outputs, remote access, and streamlining of services and operations, all of which can be obtained by operating the system digitally instead of physically without any additional material or investment cost. It also gave a glimpse of the future of different industries with DT as one of their core technologies.

DT as a technology is in its infancy, meaning it is far from reaching its full potential. Identifying and addressing the challenges of DT is crucial in leveraging the technology in different sectors. A lot of challenges associated with DT can be attributed to the novelty of the technology: lack of consensus on its definition and its value, lack of standards and regulations, lack of competent engineers and technicians, and lack of supporting software. There are issues of data security and ownership associated with DT that demand more attention since data serve as the foundation for DT. In addition to addressing the present challenges related to DT, there is also a need to broaden our horizons and identify the problems that DT as a technology can face or create in the future. Understanding the holistic view on DT—which involves what qualifies as DT, its characteristics, its advantages, how can it be implemented, and its challenges—is essential if we want to unleash the true potential of the technology.

**Author Contributions:** Conceptualization, M.S. and E.F.; writing—original draft preparation, M.S.; writing—review and editing, E.F. and E.P.H.; writing—review, Y.Q. and D.D.; supervision, E.F., D.D., and N.M. All authors have read and agreed to the published version of the manuscript.

**Funding:** This research was funded by grant number 16/RC/3918 and the APC was funded by Science Foundation Ireland.

**Institutional Review Board Statement:** Not Applicable.

**Informed Consent Statement:** Not Applicable.

**Conflicts of Interest:** The authors declare no conflict of interest.

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
