# Peer review of "Digital Twin: Origin to Future"

_asi, doi:10.3390/asi4020036_

Round 1
Reviewer 1 Report
This is probably one of the most comprehensive overview and compilation of concepts and paradigms related to Digital Twin, and one that is as informative as it can be without getting into technical details or domain specific applications. For that reason the paper is obviously a worthy publication as it is. I cannot think of any changes to suggest, except for very minor typos, e.g. "...or can(NOT) make new predictions ..." line 150).
The only point that can be criticised is that the authors did not attempt to synthesize the information in a form that shows the critical aspects of DTs and DT implementation. My only suggestion (for follow-up publication) would therefore be to capitalise on the information compiled in this paper and provide a DT descriptive framework, conceptual architecture, implementation guideline/roadmap, or even a mapping between DT features and application examples/business benefits.
Reviewer 2 Report
The paper is globally well written and structured! However, small details must be reviewed:
- Some occurrences of "Dt" instead of "DT"..."Digital twin" or "Digital Twin"? please review and try to be coherent!
- The dot "." must appear at the end of the paragraph, after references...example "on ‘Digital Twin’. (See Section 3.3 for Digital Model’ and ‘Digital Shadow’)"
- The sentence is not clear..."...there are two types of DT based on when is it developed during the life-cycle of the product...", could you, please, clarify better?
- Page 10 has a considerable white space. Suggestion: reduce the size of Figure 6 or, pass some text that is after figure 6 to previous page (page 10);
It would be interesting to describe the way the literature was selected, in the sources consulted and in the criteria used. A short paragraph about that should be relevant in the paper.
Reviewer 3 Report
The paper addresses an important gap in the states of the art and practice, as it is the case of the Digital Twin term and the many analogous ones in its surroundings. The authors do this in a theoretical way, which is perfectly fine given that this is a review paper.
The paper presents a significant analysis of the existing literature, and it does it the proper way given that opinions are emitted, viewpoints are to a fine point criticized and the own opinion of the authors is clearly stated. This is well-done and should be done this way in the case of a review paper.
I believe the paper adds theoretical value to the field, it eases the differentiation and understanding of the many Digital Twin-related terms and set important views for the future of these.
I therefore also believe the paper can be published in its present form, it is also well-structured, follows the MDPI formatting standards pretty well, and again, adds some value to the journal and the field.
Author Response
Thank you so much for your valuable suggestions and comments. We appreciate you taking the time for reviewing the paper so that the paper can be improved.
Reviewer 4 Report
This is a comprehensive and well written review paper. The topic of Digital Twins (DTs) is interesting and the paper provides a general overview, classification and challenges for the field. The focus is mostly on manufacturing DTs. I liked the explicit recognition of the "human-in-the-loop", as many predecessors to DTs have it (e.g., particularly in safety systems - aircraft autopilots, driver assistant systems in cars) and DTs will have to have humans in the loop in the future.
My concern is that DTs are not entirely new, as researchers in many fields have addressed "real-time models" (with or without data assimilation) and automation in their work. Therefore, the novelty of the term "Digital Twins" does not imply the novelty of the entire concept. Section 4.2.1 focuses on the novelty of technology, but I would've liked to see a bit of analysis/criticism in the paper of the people jumping on the bandwagon.
The other criticism of this work is in the lack of exploration of bi-directional "transfer/sharing of data". The way I saw it, but it wasn't explicitly addressed in the paper, is why is it bi-directional. I guess, that is because of automation, such that the setting/operation of the physical twin can change based on the digital counterpart. This topic is unexplored in the paper, but requires a bit more attention as it is probably the only addition to "on-line modelling" or "real-time modelling" as one-directional linking of the digital and physical counterparts is called in many fields. However, if automation is the main reason for the bi-directional data exchange, the DT should also have an optimisation model (in addition to the simulation model) to improve the design or operation of the system being modelled. This can also be an interesting topic for discussions in the paper. Following on, I'm not completely sure that the definition should state that "It is not just the digital twin which mimics its physical twin but any changes in the digital twin are mimicked by the physical twin too.”
Consequently, I don't fully agree with the statement that DT as a technology is in its infancy. Many elements of the technology are available and the authors have not identified anything that has not been used in practice.
Reviewer 5 Report
The paper reviews the Digital Twin technology application in different sectors, which make the purpose of the paper quite wide. Hence certain parts of the paper become rather descriptive and others quite short. In general the paper should be enriched with examples
I suggest the following improvements:
- In the first sentence of the Abstract you give the definition of the DT. Please try to add immediately what is the purpose of making the DT, before saying that "its implementation it is expected to grow..."
2. In the introduction it is necessary again to state the purpose of make a DT. You already state types of DTs in 3.3.3 but it is important to make mention them briefly here in the introduction.
3. You mention that the essence of the DT is the bidirectional communication between the physical and virtual model, which is true. However, you do not explain anywhere in the paper how these data are collected, what type of data are collected, the nature of these data , what can be used what cannot be used. Mentioning just IoT is very generic. Please try to elaborate a section on this aspect. This can clarify the purpose of the DTs that you describe.
4. In connection to point 3, you state advantages of the Digital Twin, which according to you are Speed of prototyping and Cost effective. Do not confuse Digital modelling CAD, CAE, BIM, etc. with DT. Making a perfect DT with bidirectional connection as you are proposing in reality it is time consuming and not always necessary and not always cost effective, since not all the necessary it is completely available. This is typical of emerging technologies. Please try to clarify this aspect .
5. Regarding the references I would have preferred that you cite more recent ones, since you yourself say that there is an exponential increase in publications related to DT. Therefore the concept is more under strong developement. A lot of references from 2016 are already quite old and their concepts overpassed.
6. Finally, try to extend the section 3.3.3 with actual practical application of DT, which can make the types of DT that you mention more clear. For example these are some works of the DT application in the Architecture, Engineering and Construction Industry:
Liu, Zhansheng, et al. "Intelligent Tensioning Method for Prestressed Cables Based on Digital Twins and Artificial Intelligence." Sensors 20.24 (2020): 7006.
Angjeliu, et al. "Development of the simulation model for Digital Twin applications in historical masonry buildings: The integration between numerical and experimental reality." Computers & Structures 238 (2020): 106282.
Alizadehsalehi, et al. "Digital twin-based progress monitoring management model through reality capture to extended reality technologies (DRX)." Smart and Sustainable Built Environment (2021).
Mechanical Engineering and software commercial products:
https://www.ansys.com/products/digital-twin
Of course try to make a more thorough research because there are many other examples in other industry sectors. You should include them in order to clarify different types of DTs that you mention.
Round 2
Reviewer 5 Report
As noted in the previous review, the paper lacks of practical applications, and becomes rather general. The authors do not state any real application of digital twin. Hence, in the present form the work becomes merely a fancy use of the word "Digital Twin" rather than a scientific paper.
Please address the elements suggested in the previous review in order to seriously improve the paper.